# Kinesin-2 autoinhibition requires elbow phosphorylation

**Guanghan Chen[1,2,3,4], Zhengyang Guo[1,2,3,4], Zhiwen Zhu[5], Shanshan Xie[6], Tianhua Zhou[7,8], Guangshuo Ou[1,2,3,4]***

[1]Tsinghua-Peking Center for Life Sciences, Tsinghua University, Beijing, China; [2]Beijing Frontier Research Center for Biological Structure, Tsinghua University, Beijing, China; [3]McGovern Institute for Brain Research, Tsinghua University, Beijing, China; [4]School of Life Sciences, Tsinghua University, Beijing, China; [5]Institute of Molecular Enzymology, Soochow University, Suzhou, China; [6]Children's Hospital, National Clinical Research Center for Child Health, Zhejiang University School of Medicine, Hangzhou, China; [7]State Key Laboratory for Membrane Biology, Beijing, China; [8]Department of Cell Biology, Zhejiang University School of Medicine, Hangzhou, China

***For correspondence:**
guangshuoou@mail.tsinghua.edu.cn

**Competing interest:** The authors declare that no competing interests exist.

## eLife Assessment

In their **important** manuscript, Chen et al. investigate the phospho-regulation of the *C. elegans* kinesin-2 motor protein OSM-3, revealing that the kinase, NEKL-3, phosphorylates a serine/threonine patch at the hinge region of the motor to mediate autoinhibition until it reaches the ciliary middle segment. The findings are supported by robust genetic data, in vivo imaging, and motility assays with wild-type and mutant motors. Overall, the study provides a **compelling** contribution to understanding the regulation of OSM-3 kinesin activity both on the molecular and cellular levels.

**Abstract** Kinesin motor proteins facilitate microtubule-based transport by converting chemical energy into mechanical forces, but this activity is autoinhibited until cargo is loaded. Regulatory mechanisms underlying this autoinhibitory conformation are not well understood. Here, we show that a NEver in mitosis Kinase NEKL-3 directly phosphorylates a flexible elbow region between two coiled-coil domains connecting the motor head and tail of an intraflagellar transport kinesin, OSM-3. The phosphor-dead (PD) mutation, but not phosphor-mimic (PM) mutation, induces constitutive motility of OSM-3 in vitro. Using knock-in animals, we discovered that both PD and PM mutations shorten the *C. elegans* sensory cilia. The constitutively active OSM-3PD fails to enter cilia and abnormally accumulates in neurites, mimicking another hyperactive mutation, OSM-3G444E. Conversely, OSM-3PM enters cilia but moves at a reduced speed, indicating an inhibitory role of elbow phosphorylation in kinesin motility. These findings highlight the crucial role of elbow phosphorylation in regulating kinesin autoinhibition.

## Introduction

Kinesin motor proteins, integral to intracellular transport along microtubules, are tightly regulated for precise cargo delivery within cells (*Burute and Kapitein, 2019*; *Christensen and Reck-Peterson, 2022*; *Hirokawa et al., 2009*; *Ou and Scholey, 2022*). Dysregulation of kinesin activity is linked to neurodegenerative disorders like Alzheimer's and Parkinson's diseases (*Brady and Morfini, 2017*; *Sleigh et al., 2019*). Understanding these regulatory mechanisms is vital for elucidating disease mechanisms and developing potential therapeutic strategies. Mechanisms controlling kinesin activation involve protein interactions, post-translational modifications, and cargo binding (*Cason and*

*Holzbaur, 2022*; *Verhey and Hammond, 2009*). Protein phosphorylation is recognized for its regulatory role in a variety of proteins, yet its impacts on kinesin motility are not fully understood (*Banerjee et al., 2021*; *DeBerg et al., 2013*; *Espeut et al., 2008*; *Liang et al., 2014*; *Sato-Yoshitake et al., 1992*; *Schäfer et al., 2008*).

Cilia serve as a unique model for studying how microtubule-based motor proteins are regulated in specific cellular regions (*Anvarian et al., 2019*; *Nachury and Mick, 2019*). The formation of cilia relies on bidirectional intraflagellar transport (IFT) along microtubules within the axoneme (*Ishikawa and Marshall, 2011*; *Klena and Pigino, 2022*; *Taschner and Lorentzen, 2016*). At the ciliary base, kinesin-2 family proteins undergo conformational changes to transport IFT particles loaded with ciliary precursors to the axonemal tip (*Ou and Scholey, 2022*; *Taschner and Lorentzen, 2016*). After unloading cargo, kinesin-2 is deactivated, while dynein-2 is activated to recycle the anterograde IFT machinery (*Ou and Scholey, 2022*; *Prevo et al., 2017*; *Taschner and Lorentzen, 2016*). In *Caenorhabditis elegans*, kinesin-2 and OSM-3 collaborate to construct sensory cilia in chemosensory neurons (*Ou et al., 2005*; *Ou and Scholey, 2022*; *Prevo et al., 2015*). Initially, kinesin-2 transports IFT particles, building the middle ciliary segments, while OSM-3 is inactive and transported by kinesin-2 (*Mitra et al., 2024*; *Ou et al., 2005*; *Ou and Scholey, 2022*; *Prevo et al., 2015*). Later, kinesin-2 transfers IFT particles to OSM-3, activating it to convey cargo molecules and assemble the distal axoneme (*Mitra et al., 2024*; *Ou et al., 2005*; *Ou and Scholey, 2022*; *Prevo et al., 2015*). The molecular mechanisms that govern the activation and inactivation of kinesin-2 and dynein-2 along the cilia are still unclear. Additionally, the regulation of OSM-3's regional motility presents an even greater mystery: Deleting OSM-3 only disrupts the distal axoneme without affecting the ciliary middle region (*Mitra et al., 2024*; *Ou et al., 2005*; *Ou and Scholey, 2022*; *Prevo et al., 2015*). It remains unclear how OSM-3 becomes enriched in the distal domain than the middle segment.

The 'elbow' of a kinesin refers to a flexible region situated between two coiled-coil domains, linking the motor head and tail of the kinesin protein (*Tan et al., 2023*; *Weijman et al., 2022*). This junction imparts necessary flexibility and facilitates interactions between the globular motor head and the rigid stalk-like tail (*Tan et al., 2023*; *Verhey and Hammond, 2009*; *Weijman et al., 2022*). The elbow region is thus considered to serve as a pivotal regulatory center, governing the motor's activity and facilitating precise spatiotemporal control of intracellular transport processes (*Tan et al., 2023*; *Weijman et al., 2022*). However, the post-translational modification of the kinesin elbow and its physiological consequences have not been elucidated. In our previous study, a genetic screen targeting the OSM-3(G444E) hyperactive mutation identified NEKL-4, a member of the NIMA kinase family, as a suppressor of this phenotype (*Yi et al., 2018*). This finding, combined with reports that NIMA kinases regulate ciliary processes independently of their canonical mitotic roles (*Chivukula et al., 2020*; *Fry et al., 2012*; *Smith et al., 2006*; *Thiel et al., 2011*), prompted us to investigate whether NIMA kinases modulate OSM-3-driven IFT. We hypothesized that NEKL-3/4, as paralogs within this family, might directly phosphorylate OSM-3 to regulate its motility. In this study, we demonstrate that the NEKL-3 kinase directly phosphorylates the elbow of OSM-3 kinesin. Our in vitro and in vivo findings suggest that elbow phosphorylation exerts an inhibitory effect on OSM-3 motility. We show that elbow phosphorylation inhibits OSM-3 motility in the soma and dendrites of sensory neurons, while its dephosphorylation is necessary for activating OSM-3 in sensory cilia.

## Results

### NEKL-3 phosphorylates the OSM-3 kinesin's elbow

To determine whether NIMA kinase family members could directly phosphorylate OSM-3, we purified prokaryotic recombinant *C. elegans* NEKL-3/NEKL-4 and OSM-3 protein in order to perform in vitro phosphorylation assays. We were able to obtain active recombinant NEKL-3 but not NEKL-4. The in vitro phosphorylation assays showed that NEKL-3 directly phosphorylates OSM-3 (*Figure 1A, B*, *Supplementary file 1*). Subsequent mass spectrometric analysis revealed phosphorylation at residues 487–490, which localize to the conserved 'YSTT' motif within OSM-3's C-terminal tail region (*Figure 1A–E*, *Figure 1—figure supplement 1*, *Supplementary file 1*). To gain structural insights from this motif, we employed LocalColabFold based on AlphaFold2 to predict the dimeric structure of OSM-3 (*Evans et al., 2022*; *Jumper et al., 2021*; *Mirdita et al., 2022*). The highest confidence model was selected for further analysis (*Figure 1C*, *Figure 1—figure supplement 2A, B*). These four

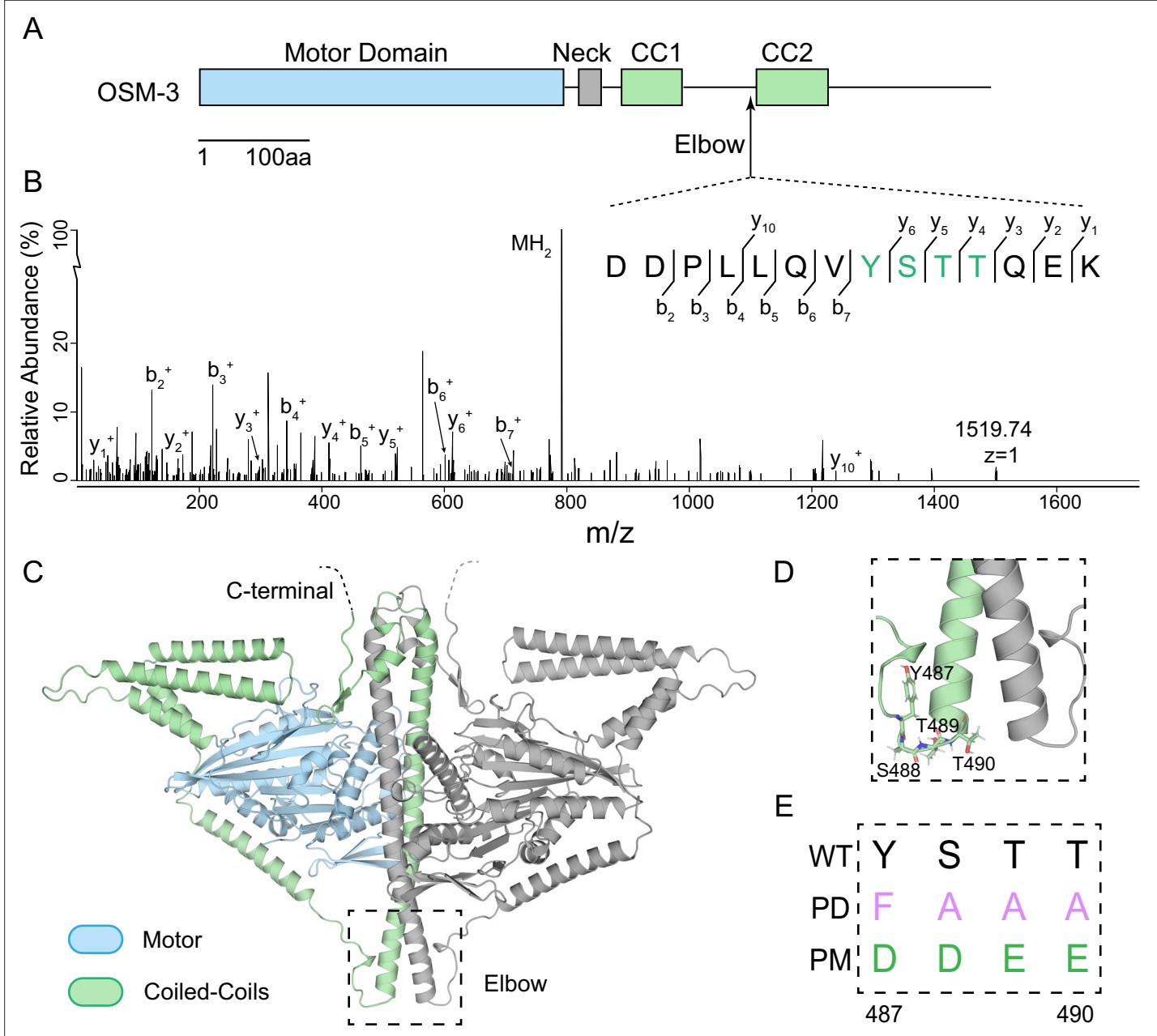

**Figure 1.** NEKL-3 phosphorylates OSM-3 at its 'elbow'. (**A**) Schematic of the full-length OSM-3. Motor domain (blue), neck (gray), and coiled-coils (green) are indicated. CC, coiled-coil. (**B**) Mass spectrum of an OSM-3 peptide that was phosphorylated by NEKL-3. Phosphorylated gel bands were subjected to MS analysis searching for phosphorylation modifications. Residues 487–490 of OSM-3 were phosphorylated and were marked by green color. (**C**) Phosphorylated residues 488–490 are at the 'elbow' of OSM-3. It shows the overall structure of the homodimeric OSM-3 predicted by AlphaFold2. The dashed square marks the 'elbow' region and is zoomed-in in **D**. (**E**) Genome-editing constructs of the elbow, showing the PD (phosphor-dead) and PM (phosphor-mimic) sequences comparing to wild type. Residues 487–490 of OSM-3 were edited to 'FAAA' for PD strain or edited to 'DDEE' for PM strain. Abbreviations: Y, Tyr; S, Ser; T, Thr; F, Phe; A, Ala; D, Asp; E, Glu.

The online version of this article includes the following figure supplement(s) for figure 1:

**Figure supplement 1.** Kinesin-2 family members have a conserved elbow motif.

**Figure supplement 2.** Predicted aligned error (PAE) and predicted Local Distance Difference Test (pLDDT) of OSM-3 dimer models.

phosphorylated 'YSTT' residues are situated at the end of the hinge2 region, linking coiled-coil1 and coiled-coil2 (*Figure 1A, C, D*). In kinesin-1, this inter-coiled-coil junction is known as the 'elbow', essential for adopting a compact conformation for autoinhibition (*Tan et al., 2023*; *Weijman et al., 2022*). Accordingly, we have designated the region comprising amino acids 487–490 as the 'elbow' of OSM-3. Although the elbow region is prevalent throughout the kinesin superfamily, the molecular regulation of the elbow conformation remains to be illustrated.

## Phosphorylation at the elbow inhibits the motility of the OSM-3 kinesin in *C. elegans*

To investigate the in vivo effects of phosphorylation and dephosphorylation at the 'elbow' region of OSM-3, we utilized genome-editing techniques to generate knock-in worms harboring phospho-dead (PD) and phospho-mimic (PM) mutations. Specifically, we replaced the amino acids 487–490 YSTT with FAAA for the PD mutation and DDEE for the PM mutation in the *C. elegans* OSM-3::GFP genome (*Figure 1D, E*). The wild-type (WT) OSM-3::GFP facilitates anterograde IFT to construct the distal ciliary segments of sensory neurons, and OSM-3::GFP fluorescence localizes along the ciliary distal segment (*Figure 2A, B*). However, OSM-3PD::GFP was absent from cilia and excluded from cell bodies, instead forming bright puncta around the axons of sensory neurons (*Figure 2A, B*). By introducing an IFT marker IFT70/DYF-1::mScarlet into *osm-3pd* worms, we revealed a marked reduction in ciliary length to 2.15 ± 0.37 μm compared to WT animals of 7.74 ± 0.94 μm, consistent with the absence of OSM-3 kinesin within the cilia (*Figure 2B, C*). The shortened cilia length and the formation of abnormal puncta at the neurite tip resemble the phenotype observed in our previously characterized OSM-3G444E mutation, which disrupts autoinhibition and leads to hyperactivation of OSM-3 (*Xie et al., 2024*). Conversely, OSM-3PM::GFP fluorescence did not form puncta but localized within the sensory cilia, albeit with shortened cilia of 4.89 ± 1.14 μm and reduced IFT speed at both middle and distal segments, indicating impaired OSM-3 motility (*Figure 2B–D*, *Figure 2—figure supplement 1A–D*).

These findings indicate that OSM-3-PM is in an autoinhibited state capable of ciliary delivery, yet fails to achieve full activation due to defective dephosphorylation. This incomplete activation results in suboptimal motor function and intermediate ciliary length phenotypes (*Figure 2B, C*). In contrast, OSM-3-PD exhibits constitutive activation leading to aggregation into axonal puncta, which completely abolishes its ciliary entry capacity (*Figure 2A, B*). Our findings collectively suggest that NEKL-3-mediated phosphorylation of the OSM-3 elbow domain serves as a critical regulatory mechanism suppressing motor activity. Under physiological conditions, this phosphorylation-dependent autoinhibition likely prevents precocious activation during cytoplasmic transport, ensuring proper delivery of motor complexes to the cilium. Subsequent dephosphorylation at the ciliary base then activates OSM-3, enabling its processive movement required for IFT.

While *nekl-3* null mutants are inviable due to essential mitotic roles (*Barstead et al., 2012*), conditional knockout (cKO) of *nekl-3* in ciliated neurons revealed its critical role in regulating OSM-3 dynamics (*Huang et al., 2025*). In *nekl-3* cKO animals, OSM-3—normally enriched in the ciliary distal segment—redistributed uniformly along the cilium, concomitant with premature activation of anterograde motility in the middle ciliary region (*Huang et al., 2025*). This phenotype aligns with our model wherein NEKL-3 phosphorylation suppresses OSM-3 activity, ensuring spatiotemporal regulation of IFT.

## Structural models of the OSM-3 kinesin and its mutants

To explore the impact of the elbow phosphorylation on the OSM-3 kinesin, we first predicted a model of the OSM-3 monomer using LocalColabFold and relaxed the mutated models with PyRosetta to calculate the energy-minimized conformations of these mutations (*Figure 3A*; *Chaudhury et al., 2010*; *Evans et al., 2022*; *Jumper et al., 2021*; *Mirdita et al., 2022*). The WT and PM models resulted in similar conformations, with their tails binding to the motor head and forming a reverse β-sheet secondary structure (*Figure 3A*). The motor–tail interaction, reported in other motors, typically results in an autoinhibitory state (*Coy et al., 1999*; *Dietrich et al., 2008*; *Espeut et al., 2008*; *Friedman and Vale, 1999*). In contrast, the G444E mutation, a characterized hyperactive mutation, showed a head–tail dissociation conformation (*Figure 3A*; *Imanishi et al., 2006*; *Xie et al., 2024*).

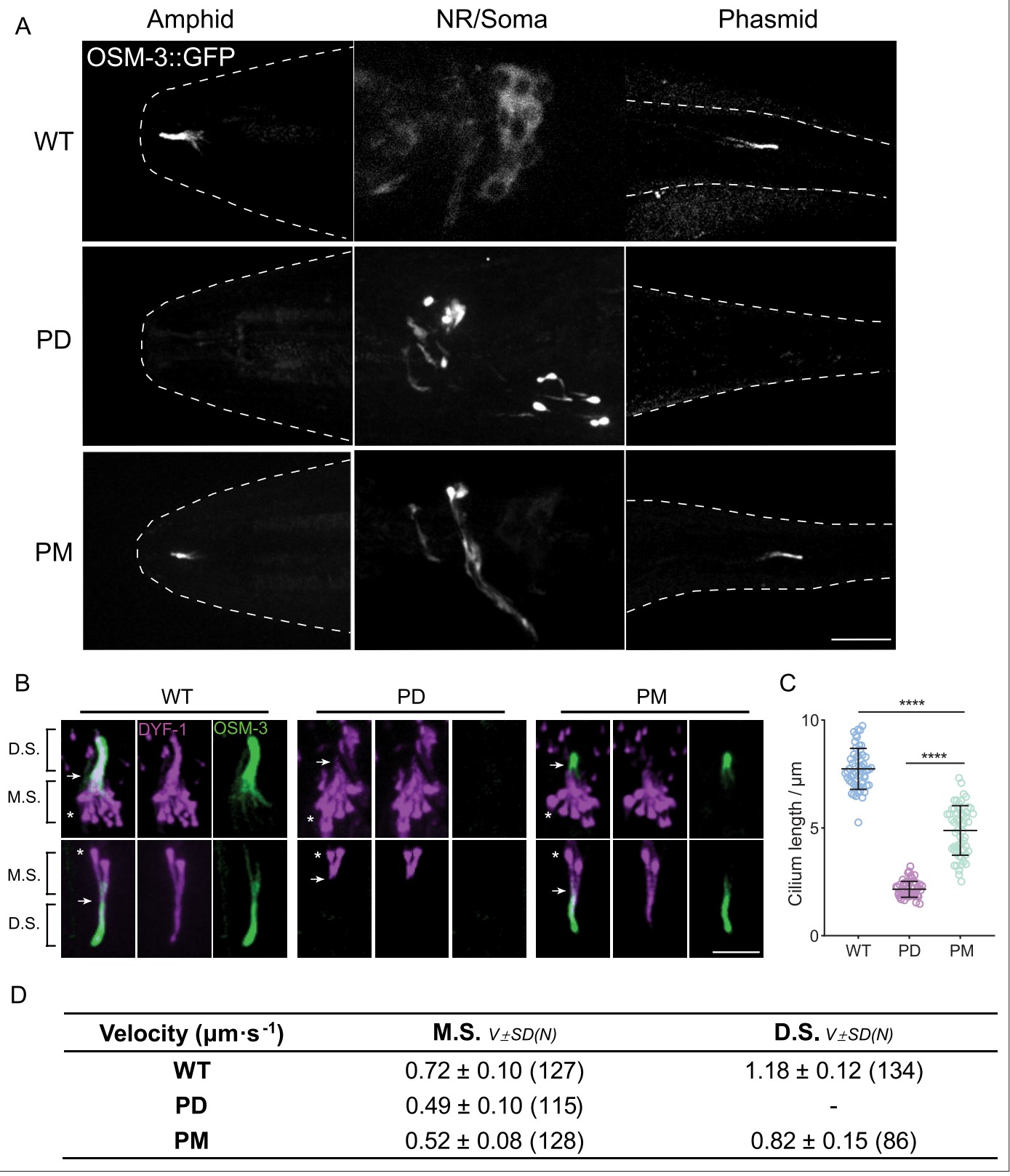

**Figure 2.** Phosphorylation at the elbow of OSM-3 is inhibitive in vivo. (**A**) Representative images of the phospho-dead (PD) and phospho-mimic (PM) knock-in worms showing their OSM-3 signal at amphid cilia, amphid neuronal soma, and phasmid cilia, respectively. The contours of the worms are marked by white dashed lines. Scale bar, 10 μm. (**B**) Representative images of the cilia from PD and PM worms marked by the ciliary marker DYF-1::mScarlet. White arrows indicate the junction between middle and distal segments, while the asterisks indicate the ciliary base. M.S., middle segment;

*Figure 2 continued on next page*

*Figure 2 continued*

D.S., distal segment. Scale bar, 5 μm. (**C**) Statistics of the cilium length of the strains shown in (**B**). The lengths of DYF-1::mScarlet signals were measured and analyzed. (**D**) Intraflagellar transport (IFT) velocities of PD and PM worms. ****p < 0.0001, analyzed by one-way ANOVA, p values were adjusted by BH method.

The online version of this article includes the following figure supplement(s) for figure 2:

**Figure supplement 1.** Statistical analysis of the intraflagellar transport (IFT) velocities in *osm-3pd* and *osm-3pm* worms, corresponding to *Figure 2D*.

Similarly, the OSM-3PD protein also exhibited a head–tail dissociation conformational change during relaxation (*Figure 3A*).

To trace the clues of the conformational changes, we evaluated the energy of pre-relaxed structures around the phosphorylated residues (*Figure 3B, C*, *Figure 3—source data 1*, *Supplementary file 2*). Compared to the PM model of OSM-3, the PD-mutated side chains had lower side-chain rotamer energy (the 'fa_dun' term) (*Alford et al., 2017*), likely due to the larger and higher charged residues in the PM state. However, the PD mutation residues exhibited higher folding/unfolding free energy (the 'ref' term) (*Alford et al., 2017*), indicating an unstable folding state of the mutated loop (*Figure 3B, C*, *Figure 3—source data 1*, *Supplementary file 2*). This instability may explain the observed conformational change leading to the autoinhibition release of OSM-3PD.

## Biochemical analyses reveal the role of elbow phosphorylation in OSM-3 motility

To directly observe the effects of phosphor mutants in the elbow region of OSM-3 in microtubule binding and motility, we examined the microtubule-stimulated ATPase activity, microtubule gliding activity, and single-molecular movement of the WT and mutant OSM-3 variants using prokaryotic recombinant proteins (*Figure 4A–F*, *Figure 4—figure supplement 1*). The WT full-length OSM-3 exhibited minimal ATPase activity (6.97% of kinesin heavy chain [KHC]) due to its autoinhibition (*Figure 4A*). As previously reported, the hyperactive OSM-3G444E mutation showed a significant increase in ATPase activity (131.65% of KHC) (*Figure 4A*, *Imanishi et al., 2006*; *Xie et al., 2024*). Consistent with our structural predictions, the OSM-3PD mutation also led to an upregulation of ATPase activity (98.15% of KHC) to a level similar to OSM-3G444E (*Figure 4A*). In contrast, OSM-3PM caused a slight increase in ATPase activity (34.28% of KHC) compared to WT but was markedly lower than OSM-3G444E or OSM-3PD (*Figure 4A*). The PM mutant's partial ATPase activity (34.28% of KHC) might arise from imperfect phosphomimicry—while the DDEE substitution introduces negative charges, it lacks the steric bulk of phosphorylated tyrosine (pY487). And this incomplete mimicry allows residual autoinhibition, sufficient to limit ciliary construction in vivo. Notably, the open-state OSM-3 mutants (e.g., G444E) displayed elevated ATPase activity, consistent with structural predictions of autoinhibition release (*Figures 3A and 4A*; *Xie et al., 2024*). While hydrodynamic profiling (e.g., SEC) could further resolve conformational states, our functional assays directly connect predicted structural changes to altered biochemical and cellular activity.

Next, we performed microtubule gliding assays to examine the gliding activity of OSM-3 and its mutants. In support of the changes in ATPase activities, OSM-3PD exhibited increased gliding speed (0.67 ± 0.08 μm/s) compared to WT (0.52 ± 0.06 μm/s) but was slower than OSM-3G444E (0.87 ± 0.09 μm/s) (*Figure 4B–D*). To examine the processivity of OSM-3, we used a total internal reflection fluorescence microscope to visualize GFP-labeled motors. We did not detect any processive movement of WT OSM-3 or OSM-3PM on microtubules (*Figure 4B, E–G*). In contrast, OSM-3PD, similar to the hyperactive OSM-3G444E, underwent persistent movement along microtubules, albeit at a slower speed and with a lower landing rate (*Figure 4B, E–G*). These data indicate that the OSM-3PD mutation disrupts the autoinhibition of full-length OSM-3, providing in vitro evidence for the inhibitory effect of NEKL-3 phosphorylation at the OSM-3 elbow region in regulating its autoinhibition.

## Genetic suppressor screens identified an intragenic suppressor of OSM-3PD

We performed a forward genetic suppressor screen to identify mutations that could restore the ciliary function of *osm-3PD* (*Figure 5A*). After screening over 10,000 haploid genomes, we isolated two suppressor mutants that exhibited the uptake of the fluorescent dye DiI through cilia, similar to WT

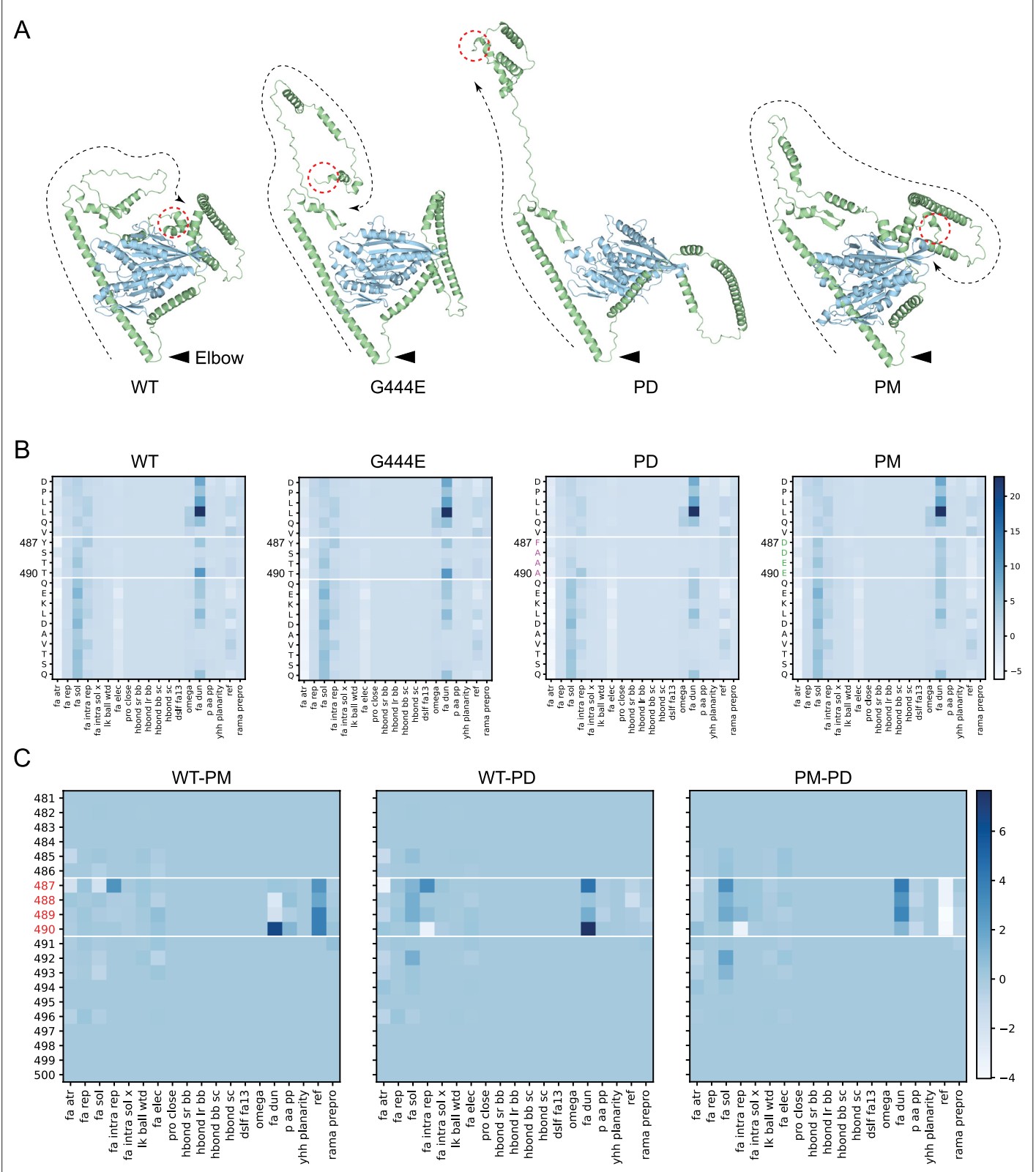

**Figure 3.** Structural models of the OSM-3 kinesin and its mutants. (**A**) Relaxed structure models of OSM-3 and mutants. Black arrowheads indicate the elbow while red dashed circles mark the C-terminus of the protein. Black dashed lines showed the extending direction from the elbow toward the C-terminus. Wild-type (WT) and phosphor-mimic (PM) showed close interaction between the tail and motor domain, while G444E and phosphor-dead (PD) showed that the tails are far away from the motor. (**B**) Heatmaps of the energy states of the pre-relaxed structure models from amino acid 481st to 500th,

*Figure 3 continued on next page*

Figure 3 continued

as labeled on the left; the amino acids between the white lines are the elbow region; each row represents an energy item as labeled on the bottom. (C) Heatmaps comparing the energy states by direct subtraction between the mutants and WT. The PD mutant has lower 'fa_dun' energy while having higher 'ref' energy than that of the PM mutant. Energy terms are explained in *Figure 3—source data 1*.

The online version of this article includes the following source data for figure 3:

**Source data 1.**

animals (*n* = 100 for each) (*Figure 5B, C*). Both suppressors were intragenic and led to the same A489T missense mutation in the elbow region (*Figure 5B, C*). Whole-genome sequencing data indicated that these two strains carried distinct mutations in many other loci (*Supplementary files 3 and 4*), supporting the notion that these two alleles were isolated independently from our suppressor screens.

We studied the in vivo behaviors of the OSM-3PD-A489T variant and observed that the vast majority of GFP fluorescence was localized within cilia (*Figure 5D*). Notably, neither of the GFP-tagged OSM-3PD-A489T mutants displayed any discernible GFP puncta throughout the entirety of the *C. elegans* body (*N* > 100 for each double mutant), indicating that the A489T mutation completely eliminated the aberrant accumulation of OSM-3PD (*Figure 5D*). By measuring IFT speeds, we showed that OSM-3PD-A489T moved indistinguishably from WT along the middle and distal ciliary segments (*Figure 5D, E*, *Figure 5—figure supplement 1A–D*). These observations were further confirmed in transgenic animals expressing GFP-tagged OSM-3PD-A489T in the *osm-3(p802)* null allele background (*Figure 5D, E*, *Figure 5—figure supplement 1A–D*).

To probe the regulatory role of T489 phosphorylation, we generated *osm-3(T489E)* and osm-*3(T489A)* mutant animals. Strikingly, both mutants formed axonal puncta (*Figure 5—figure supplement 2A–C*), recapitulating the hyperactive phenotype of the OSM-3G444E mutant. While the similar puncta formation in T489E and T489A mutants initially appeared paradoxical, this observation underscores the necessity of dynamic phosphorylation cycling at T489 for proper autoinhibition. Specifically, the T489A mutant likely disrupts phosphorylation-dependent autoinhibition stabilization, leading to constitutive activation, whereas the T489E mutant may mimic a 'locked' phosphorylated state, preventing dephosphorylation-dependent release of autoinhibition in cilia and trapping OSM-3 in an aggregation-prone conformation. These results highlight T489 as a structural linchpin whose post-translational modification dynamically regulates motor activity. While the precise molecular mechanism—such as how phosphorylation modulates tail–motor domain interactions—remains to be elucidated, our data conclusively demonstrate that perturbing T489 (even in isolation) destabilizes autoinhibition, driving puncta formation and the constitutive activity.

Collectively, these in vivo findings indicate that the A489T mutation suppresses the defects of OSM-3PD, restoring its normal function and localization.

## Discussion

This study introduces a model that elucidates the essential role of OSM-3 elbow phosphorylation in modulating kinesin motility (*Figure 6*). We demonstrate that NEKL-3 kinase phosphorylates the elbow region of OSM-3 kinesin in vitro. Fluorescence of OSM-3::GFP and NEKL-3::GFP was observed in the soma and dendrites of sensory neurons, indicating the sites of protein synthesis and potential regulatory activity. We propose that OSM-3 is synthesized in the soma, where its elbow region is subsequently phosphorylated by NEKL-3. This phosphorylation is critical to inhibit OSM-3 motility prior to its arrival at the cilia. Previous findings have shown that cytoplasmic dynein-1 facilitates the transport of centrioles from the soma along the dendrites to the dendritic tip, the site of ciliary base formation (*Li et al., 2017*). We hypothesize that inactive OSM-3 is similarly transported, possibly by dynein-1, to the ciliary base as a cargo molecule. Dephosphorylation at the ciliary middle segment appears necessary to activate OSM-3, thereby driving IFT along the ciliary distal segments.

In the absence of phosphorylation at the soma or dendrites, as demonstrated in the OSM-3PD model, OSM-3 may remain constitutively active. This constitutive activity might induce cellular responses that erroneously direct active OSM-3 to neurite tips rather than to cilia. Furthermore, OSM-3 harboring the G444E mutation within the hinge region exhibits constitutive activity (*Imanishi et al., 2006*; *Xie et al., 2024*). Our recent study revealed that this hyperactive form of OSM-3 was

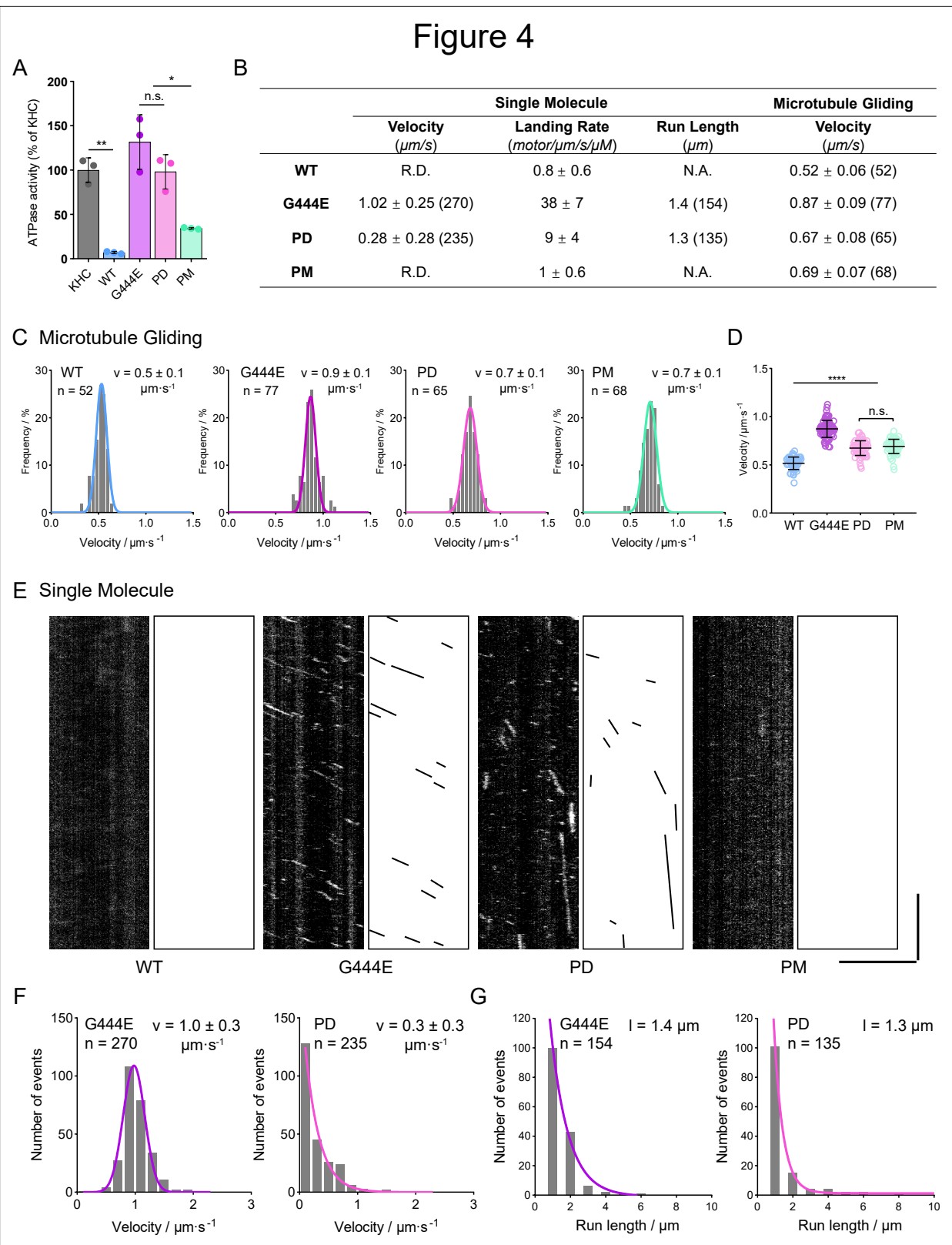

Figure 4. Phospho-dead OSM-3 behaves constitutively active in vitro while phospho-mimic OSM-3 stays autoinhibited. (**A**) Microtubule-stimulated ATPase activity of wild-type (WT) OSM-3 and mutants. G444E, the hyperactive positive control; KHC, kinesin heavy chain. Average activity of KHC was set to 100% and others were normalized to KHC. (**B**) Summary of the single-molecular assay and the microtubule gliding assay. R.D., rarely detected. N.A., not available. Data are [mean ± SD (number of events)]. (**C**) Velocity distributions of microtubule gliding assays of the indicated OSM-

*Figure 4 continued on next page*

*Figure 4 continued*

3 constructs. *n*, total events measured. *v*, µm s$^{-1}$, average velocity with standard deviation. (**D**) Statistics of microtubule gliding velocities shown in (**C**). (**E**) Representative kymographs of the single-molecular movements of WT OSM-3 and mutants as indicated. Scale bars, vertical, 10 s; horizontal, 5 µm. (**F**) Velocity distributions of the single-molecular assays. *n*, total events measured. *v*, µm s$^{-1}$, average velocity with standard deviation. The distribution of G444E was fitted with a Gaussian distribution curve, while the distribution of phosphor-dead (PD) was fitted with a one-phase decay curve. (**G**) Run length distributions of the single-molecular assays. *n*, total events measured. *l*, average run length. The curves were fitted with the one-phase decay distribution. *p < 0.05, **p < 0.01, ****p < 0.0001, analyzed by one-way ANOVA, p values were adjusted by BH method.

The online version of this article includes the following figure supplement(s) for figure 4:

**Figure supplement 1.** SDS–PAGE of purified recombinant OSM-3 mutants.

also absent from cilia and was instead expelled through membrane abscission at the tips of aberrant neurites (*Xie et al., 2024*). Adjacent glial cells subsequently engulf and degrade the extruded OSM-3G444E, a process mediated by the engulfment receptor CED-1 (*Xie et al., 2024*). We propose that the constitutively active OSM-3PD is subject to a similar fate, leading to ciliary defects analogous to those observed in the *osm-3* null or *osm-3G444E* alleles.

Our mass spectrometry analysis did not yield conclusive data regarding the specific residue in the elbow region undergoing phosphorylation. However, intriguingly, genetic suppressor screens revealed that the A489T mutation fully restores OSM-3PD localization to the cilia. Although direct mass spectrometric analysis from OSM-3PD-A489T animals is pending, the genetic suppression strongly implicates the role of the T489 site in regulating elbow phosphorylation. Considering the previous proteomic study of *Chlamydomonas* flagellar proteome identified protein phosphatases in cilia (*Pazour et al., 2005*), we postulate that a protein phosphatase may dephosphorylate inhibitory phosphorylations on the OSM-3 elbow. Once within the cilia, this phosphatase in the ciliary middle segment likely dephosphorylates OSM-3, relieving inhibition from both the motor domain and the elbow region, thereby facilitating OSM-3-driven IFT and contributing to the construction of ciliary distal segments.

In our gliding assays, OSM-3PM has an increased gliding speed of 0.69 ± 0.07 µm/s (*Figure 4C, D*), similar to PD mutant. PD and PM mutations are confined to the elbow region, leaving the motor head's mechanochemical properties intact. Upon tail immobilization—which releases autoinhibition—the gliding speeds reflect motor head activity. Single-molecule assays, however, directly resolve their native regulatory states: PD mutants are constitutively active, whereas PM mutants persist in an auto-inhibited state (*Figure 4E–G*). Although monomeric OSM-3 could theoretically mediate single-motor gliding, the previous size-exclusion chromatography coupled with multi-angle light scattering (SEC-MALS) data demonstrate that OSM-3 purifies as stable dimers (*Xie et al., 2024*). Thus, dimeric OSM-3 is perhaps the predominant functional species in our assays.

As demonstrated in our recent study (*Huang et al., 2025*), phosphorylation of OSM-3 by NEKL-3 at two distinct regions—Ser96 and the conserved 'elbow' motif—differentially regulates its activity and localization. Phosphorylation at Ser96 reduces OSM-3's ATPase activity and alters its ciliary distribution from the distal segment to a uniform localization, while elbow phosphorylation induces auto-inhibition, retaining OSM-3 in the cell body. Strikingly, in vitro phosphorylation of OSM-3 by NEKL-3 significantly reduces its microtubule-binding affinity, likely arising from combined modifications at both sites. We propose a model wherein elbow phosphorylation ensures anterograde ciliary transport, while Ser96 phosphorylation fine-tunes distal segment targeting. This multistep regulation may involve distinct phosphatases to reverse phosphorylation at specific sites, a hypothesis warranting further investigation.

It is plausible that additional kinases are involved in elbow phosphorylation. Previous studies have shown NEKL-4 can regulate the stability of ciliary microtubules by affecting tubulin glutamylation, indicating the ciliary functions of the NIMA family kinases (*Power et al., 2020*; *Power et al., 2024*). NEKL-3 and its homolog NEKL-4 exhibit similar localization patterns in the soma and dendrites. While NEKL-3 is indispensable for OSM-3 regulation, NEKL-4 appears to have a redundant role in this process. Although our phosphorylation assays utilized recombinant NEKL-3 due to technical constraints that we could not generate recombinant NEKL-4 with activity, it remains possible that NEKL-4, akin to NEKL-3, may also phosphorylate the OSM-3 elbow.

This study primarily elucidates the inhibitory phosphorylation of OSM-3 prior to ciliary entry. Equally intriguing is the regulation of OSM-3 at the ciliary tip, where its activity must be curtailed to

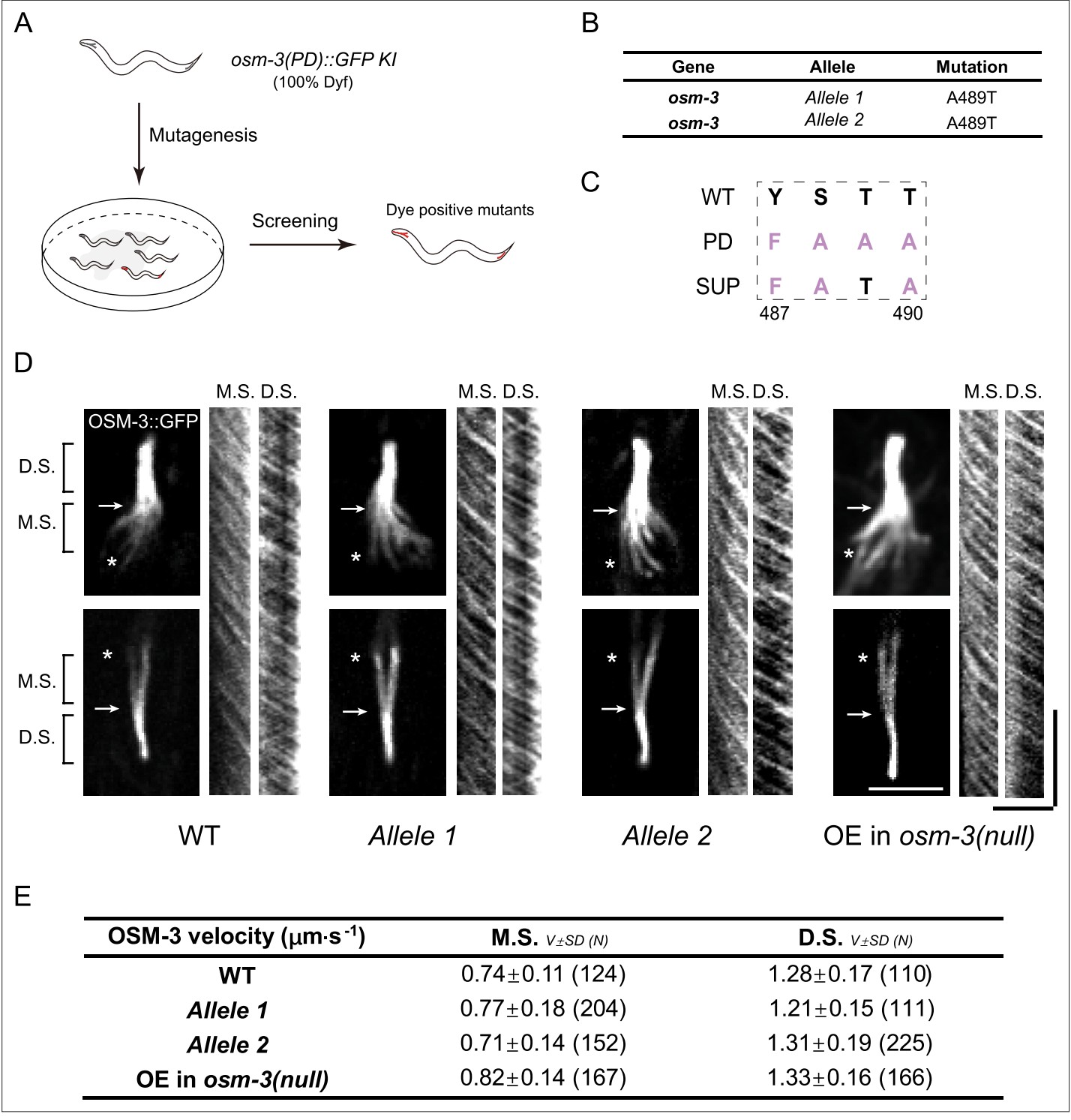

**Figure 5.** Genetic screening identified T489 as the key regulatory residue in the elbow of OSM-3. (**A**) Schematics of the forward genetic screen. 100% Dyf osm-3(PD)::GFP KI worms were mutated by ethyl methanesulfonate (EMS) and F2 progenies were screened for dye filling positive mutants. (**B**) Two independent suppressor mutants cloned from the genetic screening. (**C**) Amino acid sequences of the suppressor mutants at the elbow. (**D**) Representative images of the cilia of the suppressors and the kymographs showing the velocity of OSM-3. The rightmost panel shows the same OSM-3 version (487–489: 'FATA') with the suppressors but over-expressed under the ciliary Pdyf-1 promoter in osm-3(p802) worm. The arrows indicate the junction between middle and distal segments, while the asterisks indicate the ciliary base. M.S., middle segment; D.S., distal segment. Scale bars, vertical, 10 s; horizontal, 5 µm. (**E**) Summary of the OSM-3 velocity. M.S., middle segment. D.S. distal segment. Data are [mean ± SD (number of events)].

The online version of this article includes the following figure supplement(s) for figure 5:

*Figure 5 continued on next page*

**Figure supplement 2.** T489E and T489A of OSM-3 cause aggregated signals.

halt IFT, thus preventing excessive ciliary elongation. Previous studies have identified DYF-5, DYF-18 kinases, and other ciliary kinases as crucial regulators limiting ciliary length across various species (*Berman et al., 2003*; *Burghoorn et al., 2007*; *Maurya and Sengupta, 2021*; *Omori et al., 2010*; *Ozgül et al., 2011*; *Park et al., 2021*; *Tucker et al., 2011*; *Yi et al., 2018*). It is plausible that elbow phosphorylation by these kinases serves as a regulatory mechanism restraining OSM-3 motility at the ciliary tip. Given the ubiquity of the kinesin elbow region across the superfamily, we propose that phosphorylation regulation of this region may represent a widespread mechanism governing kinesin motility. This suggests a fundamental role for elbow phosphorylation in modulating the activity of kinesins, potentially impacting a broad array of cellular processes.

## Methods

### *C. elegans* strain culture

All *C. elegans* strains used in this study were cultured on OP50 seeded NGM (nematode growth medium) plates at 20°C. *Supplementary file 5* summarizes the strains.

### Molecular biology

The knock-in o*sm-3-pm* and *osm-3-pd* strains were generated by SunyBiotech in the *osm-3::GFP KI (SYD0199)* background by CRISPR–Cas9-based methods and confirmed by Sanger sequencing.

For OSM-3 transgenic strains, point mutation constructs were generated by PCR, and the desired constructs were co-injected with *rol-6[su1006]* into the germ line of young adult hermaphrodites. Marker-positive F1s were singled, and the F2s with transgenes were identified as transgenic lines. Two independent lines were maintained and used for experiments.

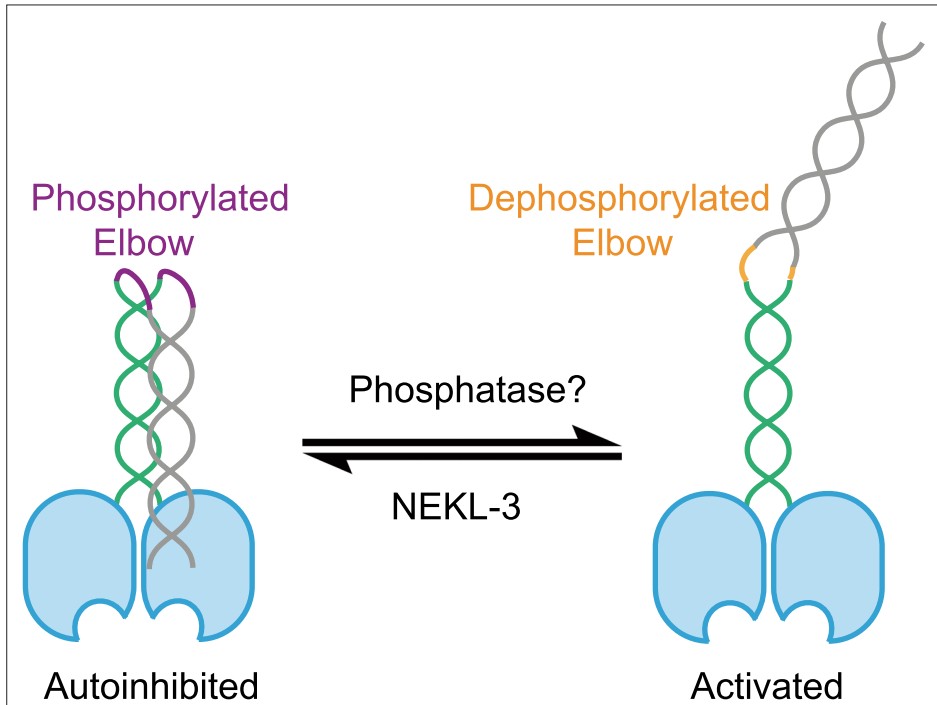

**Figure 6.** Proposed model for elbow regulation of OSM-3 by NEKL-3 phosphorylation. OSM-3 is phosphorylated at the elbow by NEKL-3 and behaves autoinhibited, while after dephosphorylation, OSM-3 turns from a compact state to an extended state due to the elbow conformational changes.

## Live imaging

Live imaging was performed as described (*Xie et al., 2024*). Basically, young adult worms were anesthetized with 0.1 mmol/l levamisole in M9 buffer and fixed on a 3% agarose pad. For the spinning disk confocal system, imaging was performed on an Axio Observer Z1 microscope (Carl Zeiss, Oberkochen, Germany) equipped with a 100×, 1.49 NA objective, an electron-multiplying charge-coupled device camera (Andor iXon+ DU-897D-C00#BV-500), and the 488/561 nm lines of a Sapphire CW CDRH USB Laser System attached to a spinning disk confocal scan head (Yokogawa CSU-X1 Spinning Disk Unit). Images were acquired by μManager (https://micro-manager.org/) at an exposure time of 200ms and analyzed with ImageJ software.

## Structure prediction and energy minimization

OSM-3 model was predicted using LocalColabFold (*Evans et al., 2022*; *Jumper et al., 2021*; *Mirdita et al., 2022*). Mutated proteins were designed by PyMOL 2.6, choosing the rotamer of the mutated residues in G444E, PM, and PD models with the least clash as the initial conformation.

To predict mutation-induced conformational changes, the initial models were subjected to Pyrosetta (*Chaudhury et al., 2010*). The energies of pre-relaxed models were evaluated with the Rosetta Energy Function 2015 (*Alford et al., 2017*), and then the relax procedure was applied to the models with default parameters to obtain the relaxed models visualized by Pymol to minimize the energy of these models. In detail, to obtain the relaxed models visualized by PyMOL and minimize the energy of these models, the classic relax mover was used in the procedure mentioned above with default settings. The relax script has been uploaded to Github: https://github.com/young55775/RosettaRelax_for_OSM3, copy archived at *GZY, 2025*.

## Protein preparation

OSM-3 was expressed in *E. coli BL21 (DE3)* and purified for in vitro assays using established protocols (*Imanishi et al., 2006*; *Xie et al., 2024*). SEC-MALS (*Xie et al., 2024*) confirmed that recombinant OSM-3 forms a homodimer (173–193 kDa) under physiological conditions, ensuring its dimeric state remained intact.

Basically, point mutations were introduced into the pET.M.3C OSM-3-eGFP-His$_6$ plasmid for prokaryotic expression. Plasmid transformed *E. coli (BL21)* was cultured at 37°C and induced overnight at 23°C with 0.2 mM IPTG. Cells were lysed in lysis buffer (50 mM NaPO$_4$ pH 8.0, 250 mM NaCl, 20 mM imidazole, 10 mM bME, 0.5 mM ATP, 1 mM MgCl$_2$, Complete Protease Inhibitor Cocktail (Roche)) and Ni-NTA beads were applied for affinity purification. After incubation, beads were washed with wash buffer (50 mM NaPO$_4$ pH 6.0, 250 mM NaCl, 10 mM bME, 0.1 mM ATP, 1 mM MgCl$_2$) and eluted with elute buffer (50 mM NaPO$_4$ pH 7.2, 250 mM NaCl, 500 mM imidazole, 10 mM bME, 0.1 mM ATP, 1 mM MgCl$_2$). Protein concentration was determined by standard Bradford assay.

*C. elegans nekl-3* cDNA was cloned into pGEX-6P GST vector and expressed in *E. coli BL21 (DE3)* and purified for in vitro phosphorylation assays. Plasmid transformed *E. coli (BL21)* was cultured at 37°C and induced overnight at 18°C with 0.5 mM IPTG. Cells were lysed in lysis buffer (50 mM NaPO$_4$ pH 8.0, 250 mM NaCl, 1 mM DTT, Complete Protease Inhibitor Cocktail (Roche)) and GST beads were applied for affinity purification. After incubation, beads were washed with wash buffer (50 mM NaPO$_4$ pH 6.0, 250 mM NaCl, 1 mM DTT) and eluted with elute buffer (50 mM NaPO$_4$ pH 7.2, 150 mM NaCl, 10 mM GSH, 1 mM DTT). Purified proteins were dialyzed against storage buffer (50 mM Tris-HCl, pH 8.0, 150 mM NaCl). Protein concentration was determined by standard Bradford assay.

## ATPase assays

Microtubule-stimulated ATPase activity assays were performed with a commercial kit (HTS Kinesin ATPase Endpoint Assay Biochem Kit, Cytoskeleton Inc) following the manufacturer's instructions.

## In vitro motility assays

Microtubule gliding assays and single-molecular assays were performed as described previously (*Imanishi et al., 2006*; *Xie et al., 2024*). Briefly, the elution peak fraction was applied to a Zeba Spin Desalting Column to exchange the protein into motility buffer (80 mM pipes-K pH 6.8, 200 mM KCl, 1 mM EGTA, 2 mM MgCl$_2$, 0.1 mM ATP, 10 mM bME) before use. For microtubule gliding assays, OSM-3 was flowed into a flow cell in the desired concentration, and rhodamine-labeled microtubules

were subsequently flowed-in with assay buffer (BRB80, 1 mM ATP/Mg$^{2+}$, 1% β-mercaptoethanol, 0.08 mg/ml glucose oxidase, 0.032 mg/ml catalase, and 80 mM glucose). For single-molecular assays, microtubules were attached to the flow cell surface via antibodies (SAP4G5), and the motors were flowed-in in a desired concentration in assay buffer. The eGFP and the rhodamine were illuminated by 488 or 561 nm laser at 20 mW, and signals were visualized by Olympus IX83 microscopy equipped with a 150× (1.45 NA, oil, Olympus) objective lens and an ORCA-Flash4.0 V3 camera. The system was controlled by Micro-Manager 2.0.

### In vitro phosphorylation assay

20 µM purified OSM-3 was incubated with 1 µM GST-NEKL-3 at 30°C in 100 µl reaction buffer (50 mM Tris-HCl pH 8.0, 10 mM MgCl$_2$, 150 mM NaCl, and 2 mM ATP) for 30 min. The reaction was terminated by boiling for 5 min with an SDS-sample buffer.

### Mass spectrometry

Following NEKL-3 treatment, OSM-3 proteins were resolved by SDS–PAGE and visualized with Coomassie Brilliant Blue staining. Protein bands corresponding to OSM-3 were excised and subjected to digestion using the following protocol: reduction with 5 mM TCEP at 56°C for 30 min; alkylation with 10 mM iodoacetamide in darkness for 45 min at room temperature, and tryptic digestion at 37°C overnight with a 1:20 enzyme-to-protein ratio. The resulting peptides were subjected to mass spectrometry analysis. Briefly, the peptides were analyzed using an UltiMate 3000 RSLCnano system coupled to an Orbitrap Fusion Lumos mass spectrometer (Thermo Fisher Scientific). We applied an in-house proteome discovery searching algorithm to search the MS/MS data against the *C. elegans* database. Phosphorylation sites were determined using the PhosphoRS algorithm with manual validation of MS/MS spectra.

### Genetic screening

To isolate suppressors of *osm-3(PD)*, L4 worms were collected in 4 ml M9 and treated with 20 µl ethyl methanesulfonate for 4 hr. Adult F2 animals were subjected to the dye-filling assay, and dye-positive mutant animals were individually cultured. Dye (DiI, 1,1′-dioctadecyl-3,3,3′,3′-tetramethylindocarbocyanine perchlorate; Sigma-Aldrich, St. Louis, MO, USA) was used at a final concentration of 20 µg/ml. Whole-genome sequencing was applied to all suppressor strains to identify candidate genes.

### Quantifications and statistical analysis

ImageJ software was used to perform measurements and quantifications of the images. For cilium length, the indicated numbers of phasmid cilia were measured. For IFT velocity, the indicated numbers of IFT particles in amphid and phasmid cilia were randomly selected for the measurement. Microtubule-stimulated ATPase activities were derived from three assays, and the average activity of KHC was set to 100%. Single molecule and gliding data were measured using ImageJ software and all the events measured were selected randomly. Statistical analyses were performed in GraphPad Prism. The statistical differences were determined by two-tailed Student's *t*-test or ANOVA analysis as described in the figure legends. The frequency distribution of IFT velocity was analyzed and fitted with a Gaussian distribution curve.

## Acknowledgements

This work was supported by the National Natural Science Foundation of China (grant no. 31991191).

## Additional information

### Funding

| Funder | Grant reference number | Author |
| --- | --- | --- |
| National Natural Science Foundation of China | 31991191 | Guangshuo Ou |

The funders had no role in study design, data collection, and interpretation, or the decision to submit the work for publication.

### Author contributions

Guanghan Chen, Conceptualization, Resources, Data curation, Formal analysis, Investigation, Writing – original draft, Writing – review and editing; Zhengyang Guo, Data curation, Software, Formal analysis, Validation; Zhiwen Zhu, Resources, Data curation; Shanshan Xie, Tianhua Zhou, Supervision, Writing – review and editing; Guangshuo Ou, Conceptualization, Supervision, Methodology, Writing – original draft, Project administration, Writing – review and editing

### Author ORCIDs

Guanghan Chen ![ORCID] https://orcid.org/0009-0009-7286-0759
Zhengyang Guo ![ORCID] https://orcid.org/0009-0001-2746-613X
Guangshuo Ou ![ORCID] https://orcid.org/0000-0003-1512-7824

Reviewer #1 (Public review): https://doi.org/10.7554/eLife.103648.3.sa1
Reviewer #2 (Public review): https://doi.org/10.7554/eLife.103648.3.sa2
Author response https://doi.org/10.7554/eLife.103648.3.sa3

## Additional files

### Supplementary files

MDAR checklist

Supplementary file 1. Mass spectrometry results of NEKL-3 treated OSM-3.

Supplementary file 2. Energy minimization results of models showed in *Figure 3*.

Supplementary file 3. WGS results of the strain GOU5380.

Supplementary file 4. WGS results of the strain GOU5381.

Supplementary file 5. *C. elegans* strains used in this study.

### Data availability

All data are available in the main text or the supplementary materials.

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
