## [Editor Report · eLife Assessment]

In their **important** manuscript, Chen et al. investigate the phospho-regulation of the *C. elegans* kinesin-2 motor protein OSM-3, revealing that the kinase, NEKL-3, phosphorylates a serine/threonine patch at the hinge region of the motor to mediate autoinhibition until it reaches the ciliary middle segment. The findings are supported by robust genetic data, in vivo imaging, and motility assays with wild-type and mutant motors. Overall, the study provides a **compelling** contribution to understanding the regulation of OSM-3 kinesin activity both on the molecular and cellular levels.

---

## [Referee Report · Reviewer #1 (Public review)]

Summary:

This manuscript is a focused investigation of the phosphor-regulation of a *C. elegans* kinesin-2 motor protein, OSM-3. In C-elegans sensory ciliary, kinesin-2 motor proteins Kinesin-II complex and OSM-3 homodimer transport IFT trains anterogradely to the ciliary tip. Kinesin-II carries OSM-3 as an inactive passenger from the ciliary base to the middle segment, where kinesin-II dissociates from IFT trains and OSM-3 gets activated and transports IFT trains to the distal segment. Therefore, activation/inactivation of OSM-3 plays an essential role in its ciliary function.

Strengths:

In this study, using mass spectrometry, the authors have shown that the NEKL-3 kinase phosphorylates a serine/threonine patch at the hinge region between coiled coils 1 and 2 of an OSM-3 dimer, referred to as the elbow region in ubiquitous kinesin-1. Phosphomimic mutants of these sites inhibit OSM-3 motility both in vitro and in vivo, suggesting that this phosphorylation is critical for the autoinhibition of the motor. Conversely, phospho-dead mutants of these sites hyperactivate OSM-3 motility in vitro and affect the localization of OSM3 in *C. elegans*. The authors also showed that Alanine to Tyrosine mutation of one of the phosphorylation rescues OS-3 function in live worms.

Weaknesses:

Collectively, this study presents evidence for the physiological role of OSM-3 elbow phosphorylation in its autoregulation, which affects ciliary localization and function of this motor. Overall, the work is well performed, and the results mostly support the conclusions of this manuscript. During revision, the authors further supported conclusions and ruled out alternative explanations by filling some logical gaps with new experimental evidence and in-text clarifications.

Comments on revisions: I have no additional comments or concerns.

---

## [Referee Report · Reviewer #2 (Public review)]

Summary:

The regulation of kinesin is fundamental to cellular morphogenesis. Previously, it has been shown that OSM-3, a kinesin required for intraflagellar transport (IFT), is regulated by autoinhibition. However, it remains totally elusive how the autoinhibition of OSM-3 is released. In this study, the authors have shown that NEKL-3 phosphorylates OSM-3 and release its autoinhibition.

The authors found NEKL-3 directly phosphorylates OSM-3 (Figure 1). The phophorylated residue is the "elbow" of OSM-3. The authors introduced phospho-dead (PD) and phospho-mimic (PM) mutations by genome editing and found that the OSM-3(PD) protein does not form cilia, and instead, accumulates to the axonal tips. The phenotype is similar to another constitutive active mutant of OSM-3, OSM-3(G444A) (Imanishi et al., 2006; Xie et al., 2024). osm-3(PM) has shorter cilia, which resembles with loss of function mutants of osm-3 (Figure 2). The authors did structural prediction and shows that G444E and PD mutations change the conformation of OSM-3 protein (Figure 3). In the single molecule assays G444E and PD mutations exhibited increased landing rate (Figure 4). By unbiased genetic screening, the authors identified a suppressor mutant of osm-3(PD), in which A489T occurs. The result confirms the importance of this residue. Based on these results, the authors suggest that NEKL-3 induces phosphorylation of the elbow domain and inactivates OSM-3 motor when the motor is synthesized in the cell body. This regulation is essential for the proper cilia formation.

Strengths:

The finding is interesting and gives new insight into how IFT motor is regulated.

Comments on revisions: In the revised manuscript, the authors describe why they focused on NEKL-3 and detailed experimental procedures are presented.

My only minor concern is the title, which appears to be too general. Researchers in the motor protein field may firstly assume this paper focuses on kinesin-1, because the "elbow" domain was originally suggested in kinesin-1. This paper newly determines the elbow region of OSM-3 and shows its crucial role in autoinhibition. Therefore, a more specific title, "Kinesin-2 Autoinhibition Requires Elbow Phosphorylation" or "OSM-3 Autoinhibition Requires Elbow phosphorylation" may be better.

---

## [Author Response]

The following is the authors’ response to the original reviews

**Public Reviews:**

**Reviewer #1 (Public review):**
Summary:This manuscript is a focused investigation of the phosphor-regulation of a *C. elegans* kinesin-2 motor protein, OSM-3. In C-elegans sensory ciliary, kinesin-2 motor proteins Kinesin-II complex and OSM-3 homodimer transport IFT trains anterogradely to the ciliary tip. Kinesin-II carries OSM-3 as an inactive passenger from the ciliary base to the middle segment, where kinesin-II dissociates from IFT trains and OSM-3 gets activated and transports IFT trains to the distal segment. Therefore, activation/inactivation of OSM-3 plays an essential role in its ciliary function.Strengths:In this study, using mass spectrometry, the authors have shown that the NEKL-3 kinase phosphorylates a serine/threonine patch at the hinge region between coiled coils 1 and 2 of an OSM-3 dimer, referred to as the elbow region in ubiquitous kinesin-1. Phosphomimic mutants of these sites inhibit OSM-3 motility both in vitro and in vivo, suggesting that this phosphorylation is critical for the autoinhibition of the motor. Conversely, phospho-dead mutants of these sites hyperactivate OSM-3 motility in vitro and affect the localization of OSM3 in *C. elegans*. The authors also showed that Alanine to Tyrosine mutation of one of the phosphorylation rescues OS-3 function in live worms.Weaknesses:Collectively, this study presents evidence for the physiological role of OSM-3 elbow phosphorylation in its autoregulation, which affects ciliary localization and function of this motor. Overall, the work is well performed, and the results mostly support the conclusions of this manuscript. However, the work will benefit from additional experiments to further support conclusions and rule out alternative explanations, filling some logical gaps with new experimental evidence and in-text clarifications, and improving writing before I can recommend publication.

We appreciate Reviewer #1’s comments and suggestions. We have now provided additional evidences and discussions to further support our conclusions and fill the logical gaps. We have also provided alternative explanations to our data and improved writing.

**Reviewer #2 (Public review):**
Summary:The regulation of kinesin is fundamental to cellular morphogenesis. Previously, it has been shown that OSM-3, a kinesin required for intraflagellar transport (IFT), is regulated by autoinhibition. However, it remains totally elusive how the autoinhibition of OSM-3 is released. In this study, the authors have shown that NEKL-3 phosphorylates OSM-3 and releases its autoinhibition.The authors found NEKL-3 directly phosphorylates OSM-3 (although the method is not described clearly) (Figure 1). The phophorylated residue is the "elbow" of OSM-3. The authors introduced phospho-dead (PD) and phospho-mimic (PM) mutations by genome editing and found that the OSM-3(PD) protein does not form cilia, and instead, accumulates to the axonal tips. The phenotype is similar to another constitutive active mutant of OSM-3, OSM-3(G444A) (Imanishi et al., 2006; Xie et al., 2024). osm-3(PM) has shorter cilia, which resembles with loss of function mutants of osm-3 (Figure 3). The authors did structural prediction and showed that G444E and PD mutations change the conformation of OSM-3 protein (Figure 3). In the single-molecule assays G444E and PD mutations exhibited increased landing rate (Figure 4). By unbiased genetic screening, the authors identified a suppressor mutant of osm-3(PD), in which A489T occurs. The result confirms the importance of this residue. Based on these results, the authors suggest that NEKL-3 induces phosphorylation of the elbow domain and inactivates OSM-3 motor when the motor is synthesized in the cell body. This regulation is essential for proper cilia formation.Strengths:The finding is interesting and gives new insight into how the IFT motor is regulated.Weaknesses:The methods section has not presented sufficient information to reproduce this study.

We appreciate that Reviewer #2 is also positive to our study. We have now provided sufficient information in the revised Methods section.

**Recommendations for the authors:**

**Reviewer #1 (Recommendations for the authors):**
Major Concerns(1) Why do the authors think that NEKL-3 phosphorylates OSM-3 in the first place? This seems to come out of nowhere and prior evidence indicating that NEKL-3 may be phosphorylating OSM-3 is not even mentioned in the Introduction.

We thank the Reviewer for raising this important point. Our hypothesis that NEKL-3 phosphorylates OSM-3 stems from prior findings in our lab. In a previous study (Yi et al., Traffic, 2018, PMID: 29655266), we identified NEKL-4, a member of the NIMA kinase family, as a suppressor of the OSM-3(G444E) hyperactive mutation. This discovery prompted us to explore the broader role of NIMA kinases in regulating OSM3. Subsequent genetic screens (Xie et al., EMBO J, 2024, PMID: 38806659) revealed that both NEKL-3 and NEKL-4 suppress multiple OSM-3 mutations, further supporting their functional interaction. Given the established role of NIMA kinases in phosphorylation-dependent processes (Fry et al., JCS, 2012, PMID: 23132929; Chivukula et al., Nat. Med., 2020, PMID: 31959991; Thiel, C. et al. Am. J. Hum. Genet. 2011, PMID: 21211617; Smith, L. A. et al., J. Am. Soc. Nephrol., 2006, PMID: 16928806), we hypothesized that NEKL-3/4 may directly phosphorylate OSM-3 to modulate its activity.

To test this hypothesis, we expressed recombinant *C. elegans* NEKL-3 and OSM-3 proteins and conducted in vitro phosphorylation assays. While we were unable to obtain active recombinant NEKL-4 (limitations noted in the revised text), our experiments with NEKL-3 revealed phosphorylation at residues 487-490 (YSTT motif) in OSM-3’s tail region, as confirmed by mass spectrometry. These findings are now explicitly contextualized in the Introduction and Results sections of the revised manuscript.

Page #4, Line #11:

“...In our previous study (Yi et al., Traffic, 2018, PMID: 29655266), a genetic screen targeting the OSM-3(G444E) hyperactive mutation identified NEKL-4, a member of the NIMA kinase family, as a suppressor of this phenotype. This finding, combined with reports that NIMA kinases regulate ciliary processes independently of their canonical mitotic roles (Fry et al., JCS, 2012, PMID: 23132929; Chivukula et al., Nat. Med., 2020, PMID: 31959991; Thiel, C. et al. Am. J. Hum. Genet. 2011, PMID: 21211617; Smith, L. A. et al., J. Am. Soc. Nephrol., 2006, PMID: 16928806), prompted us to investigate whether NIMA kinases modulate OSM-3-driven intraflagellar transport. We hypothesized that NEKL-3/4, as paralogs within this family, might directly phosphorylate OSM-3 to regulate its motility...”

Page #4, line #26:

“... To determine whether NIMA kinase family members could directly phosphorylate

OSM-3, we purified prokaryotic recombinant *C. elegans* NEKL-3/NEKL-4 and OSM3 protein in order to perform in vitro phosphorylation assays. We were able to obtain active recombinant NEKL-3 but not NEKL-4. The in vitro phosphorylation assays showed that NEKL-3, directly phosphorylates OSM-3 (Fig. 1A-B, Appendix Table S1). Subsequent mass spectrometric analysis revealed phosphorylation at residues 487-490, which localize to the conserved "YSTT" motif within OSM-3’s C-terminal tail region ...”

(2) The authors need to characterize the proteins they expressed and purified for in vitro ATPase and motility assays. Are these proteins monomers or dimers?

For our in vitro ATPase and motility assays, OSM-3 was expressed in *E. coli BL21(DE3)* and purified using established protocols (Xie et al., EMBO J, 2024, PMID: 38806659; Imanishi et al., JCB, 2006, PMID: 17000874). To confirm its oligomeric state, we analyzed recombinant OSM-3 by size-exclusion chromatography coupled with multiangle light scattering (SEC-MALS). As reported in Xie et al. (2024), OSM-3 (~80 kDa monomer) elutes with a molecular weight of 173–193 kDa under physiological buffer conditions, consistent with a homodimeric assembly. These findings confirm that the functional unit used in our assays is the biologically relevant dimer. This characterization has been added to the revised manuscript on Page #35, Line #7.

“…OSM-3 was expressed in *E. coli* BL21(DE3) and purified for in vitro assays using established protocols (REFs). Size-exclusion chromatography coupled with multiangle light scattering (SEC-MALS) (Xie et al., EMBO J., 2024) confirmed that recombinant OSM-3 forms a homodimer (173–193 kDa) under physiological conditions, ensuring its dimeric state remained intact....”

(3) The authors primarily used PD and PM mutations, which affect all four amino acids in the region. This may or may not be physiologically relevant. Figure 5 indicates that T489 is a critical regulatory site. However, this conclusion is undermined by reliance on PD mutations, which affect all four amino acids. Creating PM (T489E) and PD (T489A) mutations based on WT OSM-3 would better reflect physiological relevance. In vitro assays with a single phosphomimic or phosphor-dead mutation at residue 489 are missing at the end of this story. This would better link Figure 5 with the rest of the manuscript.

We thank the reviewer for this constructive critique. Below, we address the concerns and integrate new data to strengthen the link between T489 and autoinhibition:

To probe the regulatory role of T489 phosphorylation, we generated osm-3(T489E) (phosphomimetic, PM) and osm-3(T489A) (phospho-dead, PD) mutant animals. Strikingly, both mutants formed axonal puncta (Figure S7), recapitulating the hyperactive phenotype of the OSM-3G444E mutant. While the similar puncta formation in PM and PD mutants initially appeared paradoxical, this observation underscores the necessity of dynamic phosphorylation cycling at T489 for proper autoinhibition. Specifically, the PD mutant (T489A) likely disrupts phosphorylationdependent autoinhibition stabilization, leading to constitutive activation, where as the PM mutant (T489E) may mimic a "locked" phosphorylated state, preventing dephosphorylation-dependent release of autoinhibition in cilia and trapping OSM-3 in an aggregation-prone conformation. These results highlight T489 as a structural linchpin whose post-translational modification dynamically regulates motor activity. While the precise molecular mechanism—such as how phosphorylation modulates tailmotor domain interactions—remains to be elucidated, our data conclusively demonstrate that perturbing T489 (even in isolation) destabilizes autoinhibition, driving puncta formation and the constitutive activity.

We have integrated the above paragraph in the revised manuscript on page #8, line #27.

(4) There seems to be a disconnect between the MT gliding assays in Figure 4C and single molecule motility assays in Figure 4E. The gliding assays show that all constructs can glide microtubules at near WT speeds. Yet, the motility assays show that WT and PM cannot land or walk on MTs. The authors need to explain why this is the case. Is this because surface immobilization of kinesin from its tail disrupts autoinhibition? Alternatively, the protein preparation may include monomers that cannot be autoinhibited and cannot land and processively walk on surface-immobilized microtubules (because they only have one motor domain) but can glide microtubules when immobilized on the surface from their tail.

The surface immobilization of OSM-3 via its tail domain disrupts autoinhibition, a phenomenon previously observed in other kinesins such as kinesin-1 (Nitzsche et al, Methods Cell Biol., 2010, PMID: 20466139). In our assays, OSM-3 was nonspecifically immobilized on glass surfaces, enabling microtubule gliding by motors whose autoinhibition was relieved through tail anchoring. Critically, the PD and PM mutations reside in the tail region and do not alter the intrinsic properties of the motor head domain. Consequently, once autoinhibition is released via immobilization, the gliding velocities reflect the conserved motor head activity, which is expected to remain comparable across all constructs. While we cannot entirely rule out the presence of monomeric OSM-3 in solution, several lines of evidence argue against this possibility. First, the mutations are located in the elbow region, which is dispensable for motor dimerization. Second, SEC-MALS analysis from prior studies confirms that purified OSM-3 exists predominantly as dimers in solution.

We have discussed these issues in the revised text on page #10, line #18:

“…In our gliding assays, OSM-3PM has an increased gliding speed of 0.69 ± 0.07 μm/s (Fig. 4 C-D), similar to PD mutant. PD and PM mutations are confined to the elbow region, leaving the motor head’s mechanochemical properties intact. Upon tail immobilization—which releases autoinhibition—the gliding speeds reflect motor head activity. Single-molecule assays, however, directly resolve their native regulatory states: PD mutants are constitutively active, whereas PM mutants persist in an autoinhibited state (Fig. 4E-G). Although monomeric OSM-3 could theoretically mediate singlemotor gliding, the previous SEC-MALS data demonstrate that OSM-3 purifies as stable dimers (Xie et al., EMBO J, 2024, PMID: 38806659). Thus, dimeric OSM-3 is perhaps the predominant functional species in our assays…”

(5) An alternative explanation for the data is that both PD and PM mutations result in loss-of-function effects, disrupting OSM-3 activity. For instance:a) In Figure 2C, both mutations cause shorter cilia than the wild type (WT).b) In Figure 4A, both mutations result in higher ATPase activity than WT.c) In Figure 4D, both mutations show increased gliding velocity compared to WT. These results suggest the observed effects could stem from loss of function rather than phosphorylation-specific regulation.

Although PD and PM mutations exhibit superficially similar "loss-of-function" phenotypes in certain assays, they mechanistically disrupt motor regulation in distinct ways:

a) Ciliary Length (Figure 2C) PD Mutants: Hyperactivation causes OSM-3-PD to prematurely aggregate into axonal puncta, preventing ciliary entry. Consequently, cilia are built solely by the weaker Kinesin-II motor, which only constructs shorter middle segments.

PM Mutants: OSM-3-PM retains autoinhibition during transport (enabling ciliary entry) but cannot be dephosphorylated in cilia. This blocks activation, leaving OSM-3-PM partially functional and resulting in cilia intermediate in length between WT and PD.

We have discussed this issue in the revised text on page #5, line #30:

“…These findings indicate that OSM-3-PM is in an autoinhibited state capable of ciliary delivery, yet fails to achieve full activation due to defective dephosphorylation. This incomplete activation results in suboptimal motor function and intermediate ciliary length phenotypes (Fig.2 B-C). In contrast, OSM-3-PD exhibits constitutive activation leading to aggregation into axonal puncta, which completely abolishes its ciliary entry capacity (Fig.2 A-B)...”

b) ATPase Activity (Figure 4A)

PD Mutants: Fully autoinhibition-released (98.15% of KHC ATPase activity), consistent with constitutive activation.

PM Mutants: Show partial ATPase activity (34.28% of KHC), reflecting imperfect phosphomimicry. While the DDEE substitution introduces negative charges, it fails to fully replicate the steric/kinetic effects of phosphorylated tyrosine (Y486; phenyl ring absent), resulting in incomplete autoinhibition stabilization. Despite this, the residual inhibition is sufficient to phenocopy shorter cilia in vivo.

We have discussed this issue in the revised text on page #7, line#19:

“…The PM mutant’s partial ATPase activity (34.28% of KHC) might arise from imperfect phosphomimicry—while the DDEE substitution introduces negative charges, it lacks the steric bulk of phosphorylated tyrosine (pY487). And this incomplete mimicry allows residual autoinhibition, sufficient to limit ciliary construction in vivo...”

c) Microtubule Gliding Velocity (Figure 4D)

Gliding Assay Limitation: Tail immobilization artificially releases autoinhibition, masking regulatory differences. Thus, all constructs (PD, PM) exhibit similar velocities (~0.7 µm/s), reflecting conserved motor head activity.

Single-Molecule Assay (Figure 4E): Directly resolves native autoinhibition states:

PD mutants show robust motility (autoinhibition released).

PM mutants remain largely inactive (autoinhibition retained).

We have discussed this issue in the revised text on page #10, line#18:

“…In our gliding assays, OSM-3PM has an increased gliding speed of 0.69 ± 0.07 μm/s (Fig. 4 C-D), similar to PD mutant. PD and PM mutations are confined to the elbow region, leaving the motor head’s mechanochemical properties intact. Upon tail immobilization—which releases autoinhibition—the gliding speeds reflect motor head activity. Single-molecule assays, however, directly resolve their native regulatory states: PD mutants are constitutively active, whereas PM mutants persist in an autoinhibited state (Fig. 4E-G)...”

Minor Suggestions and Concerns(1) Lines 60-66: References that support these observations are missing from this section.

We have added the relevant references.

(2) Lines 66-67: I would revise this sentence as "It remains unclear how OSM-3 becomes enriched...".

We have made the changes.

(3) Line 85: The authors should describe how they perform these assays (i.e. recombinantly expressed NEKL-3 and OSM-3, are these *C. elegans* proteins, and which expression system was used...).

We have described them in the main text and methods

Page #4 line #26

“...To determine whether NIMA kinase family members could directly phosphorylate OSM-3, we purified prokaryotic recombinant *C. elegans* NEKL-3/NEKL-4 and OSM-3 protein in order to perform in vitro phosphorylation assays...”

Page #35 line#12

“...Basically, point mutations was introduced in to pET.M.3C OSM-3-eGFP-His6 plasmid for prokaryotic expression. Plasmid transformed *E. coli* (BL21) was cultured at 37°C and induced overnight at 23°C with 0.2 mM IPTG. Cells were lysed in lysis buffer (50 mM NaPO4 pH8.0, 250 mM NaCl, 20 mM imidazole, 10 mM bME, 0.5 mM ATP, 1 mM MgCl¬2, Complete Protease Inhibitor Cocktail (Roche)) and Ni-NTA beads were applied for affinity purification. After incubation, beads were washed with wash buffer (50 mM NaPO4 pH6.0, 250 mM NaCl, 10 mM bME, 0.1 mM ATP, 1 mM MgCl¬2) and eluted with elute buffer (50 mM NaPO4 pH7.2, 250 mM NaCl, 500 mM imidazole, 10 mM bME, 0.1 mM ATP, 1 mM MgCl¬2). Protein concentration was determined by standard Bradford assay. *C elegans* nekl-3 cDNA was cloned in to pGEX-6P GST vector and expressed in *E. coli* BL21 (DE3) and purified for in vitro phosphorylation assays. Plasmid transformed *E. coli* (BL21) was cultured at 37°C and induced overnight at 18°C with 0.5 mM IPTG. Cells were lysed in lysis buffer (50 mM NaPO4 pH8.0, 250 mM NaCl, 1 mM DTT, Complete Protease Inhibitor Cocktail (Roche)) and GST beads were applied for affinity purification. After incubation, beads were washed with wash buffer (50 mM NaPO4 pH6.0, 250 mM NaCl, 1 mM DTT) and eluted with elute buffer (50 mM NaPO4 pH7.2, 150 mM NaCl, 10 mM GSH, 1 mM DTT). Purified proteins were dialyzed against storge buffer (50 mM Tris-HCl, pH 8.0, 150 mM NaCl). Protein concentration was determined by standard Bradford assay...”

(4) Line 141: The first sentence of this paragraph lacks motivation. I would start this sentence with "To directly observe the effects of phosphor mutants in the elbow region in microtubule binding and motility of OSM-3, we...".

We have made the change.

(5) Figure 1B: The mass spectrometry data in Figure 1B lacks adequate explanation. The Methods section should detail the experimental protocol, data interpretation, and any databases used. Additionally, the manuscript should list all identified phosphorylation sites on OSM-3 to provide context, including whether Y487_T490 is the major site.

We have provided the detailed experimental protocol, data interpretation, and databases used in methods. We have provided all identified sites as Appendix table S1.

(6) Figure 1C: Is it possible to model the effect of PM and PD mutations using AlphaFold? The authors should also show PAE or pLDDT scores of their model.

AlphaFold cannot well model the effect of mutants, but we conducted the Rosetta relax to capture their possible conformational changes, as shown in the revised Figure 3. We have provided PAE and pLDDT as a new figure, Figure S2.

(7) Figure 2D: The unit for speed should use a lowercase "s" for seconds.

We have fixed it.

(8) Figure 3: I am not sure whether this figure stands for a main text figure on its own, as it is only a Rosetta prediction and is not supported by any experimental data. In addition, it remains unclear what the labels on the x-axis mean.

We have updated the figure and explain the labels on the x-axis in Figure S4 to make it more reader-friendly.

(9) Figure 4: NEKL-3-treated OSM-1 should be included as a positive control in the in vitro experiments.

We suspect that the Reviewer asked for NEKL-3-treated OSM-3.

In our other study which has just been accepted by the Journal of Cell Biology, NEKL3-treated OSM-3 significantly reduced the affinity between OSM-3 motor and microtubules and showed very low ATPase activity. We have cited and discussed this in the revised text on page #10, line #28:

“…As demonstrated in our recent study (Huang et al., JCB, 2025, In press, attached), phosphorylation of OSM-3 by NEKL-3 at two distinct regions—Ser96 and the conserved "elbow" motif—differentially regulates its activity and localization. Phosphorylation at Ser96 reduces OSM-3’s ATPase activity and alters its ciliary distribution from the distal segment to a uniform localization, while elbow phosphorylation induces autoinhibition, retaining OSM-3 in the cell body. Strikingly, in vitro phosphorylation of OSM-3 by NEKL-3 significantly reduces its microtubulebinding affinity, likely arising from combined modifications at both sites. We propose a model wherein elbow phosphorylation ensures anterograde ciliary transport, while Ser96 phosphorylation fine-tunes distal segment targeting. This multistep regulation may involve distinct phosphatases to reverse phosphorylation at specific sites, a hypothesis warranting further investigation….”

(10) Figure 4C, D, and F: The unit of velocity is wrong. The authors should use the same units they used in the table shown in Figure 4B.

We have fixed these errors

(11) Figure 4F: The velocity of PD is a lot lower than G444E. Therefore, it would be more appropriate to refer to PD as partially active, rather than hyperactive.

We have made the change.

(12) Figure 5: There is too much genetics jargon on this figure (EMF, F2, 100%Dyf,...). How are the alleles numbered? Is it OK to refer to them as Alleles 1 and 2 for simplicity?

According to the established *C. elegans* allele nomenclature, each worm allele has a unique number named after the lab code for identification. We have simplified the labels and updated the figure to make it more reader-friendly.

(13) Figure 5E: A plot would be more reader-friendly than a table. Additionally, the legend for Fig. 5E mistakenly refers to it as "D."

We have changed the table to a plot and fixed the mistakes. We thank the Reviewer for pointing them out.

**Reviewer #2 (Recommendations for the authors):**
(1) The model appears as if NEKL-3 induces dephosphorylation of OSM-3 (Figure 6). This is not consistent with the conclusions described in the Discussion and is confusing.

We have updated the model figure and fixed the error.

(2) It should be described why the authors hypothesized NEKL-3 phosphorylates OSM3. Was there genetic evidence? Did the authors screened cilia-related kinases? or Did the authors identify it incidentally? Providing this information would help readers to understand the context of the research.

We appreciate both Reviewers for pointing out this issue.

Our hypothesis that NEKL-3 phosphorylates OSM-3 stems from prior findings in our lab. In a previous study (Yi et al., Traffic, 2018, PMID: 29655266), we identified NEKL-4, a member of the NIMA kinase family, as a suppressor of the OSM-3(G444E) hyperactive mutation. This discovery prompted us to explore the broader role of NIMA kinases in regulating OSM-3. Subsequent genetic screens (Xie et al., EMBO J, 2024, PMID: 38806659) revealed that both NEKL-3 and NEKL-4 suppress multiple OSM-3 mutations, further supporting their functional interaction. Given the established role of NIMA kinases in phosphorylation-dependent processes (Fry et al., JCS, 2012, PMID: 23132929; Chivukula et al., Nat. Med., 2020, PMID: 31959991; Thiel, C. et al. Am. J. Hum. Genet. 2011, PMID: 21211617; Smith, L. A. et al., J. Am. Soc. Nephrol., 2006, PMID: 16928806), we hypothesized that NEKL-3/4 may directly phosphorylate OSM3 to modulate its activity.

To test this hypothesis, we expressed recombinant *C. elegans* NEKL-3 and OSM-3 proteins and conducted in vitro phosphorylation assays. While we were unable to obtain active recombinant NEKL-4 (limitations noted in the revised text), our experiments with NEKL-3 revealed phosphorylation at residues 487-490 (YSTT motif) in OSM-3’s tail region, as confirmed by mass spectrometry. These findings are now explicitly contextualized in the Introduction and Results sections of the revised manuscript.

Page #4, Line #11:

“... In our previous study (Yi et al., Traffic, 2018, PMID: 29655266), a genetic screen targeting the OSM-3(G444E) hyperactive mutation identified NEKL-4, a member of the NIMA kinase family, as a suppressor of this phenotype. This finding, combined with reports that NIMA kinases regulate ciliary processes independently of their canonical mitotic roles (Fry et al., JCS, 2012, PMID: 23132929; Chivukula et al., Nat. Med., 2020, PMID: 31959991; Thiel, C. et al. Am. J. Hum. Genet. 2011, PMID: 21211617; Smith, L. A. et al., J. Am. Soc. Nephrol., 2006, PMID: 16928806), prompted us to investigate whether NIMA kinases modulate OSM-3-driven intraflagellar transport. We hypothesized that NEKL-3/4, as paralogs within this family, might directly phosphorylate OSM-3 to regulate its motility...”

Page #4, line #26:

“... To determine whether NIMA kinase family members could directly phosphorylate OSM-3, we purified prokaryotic recombinant *C. elegans* NEKL-3/NEKL-4 and OSM3 protein in order to perform in vitro phosphorylation assays. We were able to obtain active recombinant NEKL-3 but not NEKL-4. The in vitro phosphorylation assays showed that NEKL-3, directly phosphorylates OSM-3 (Fig. 1A-B, Appendix Table S1). Subsequent mass spectrometric analysis revealed phosphorylation at residues 487-490, which localize to the conserved "YSTT" motif within OSM-3’s C-terminal tail region...”

(3) It is curious the authors have not addressed the cilia phenotype and the localization of OSM-3 in nekl-3 mutant. Regardless of whether these observations agrees with the proposed mechanisms, it is essential for the authors to show and discuss the cilia phenotype and OSM-3 localization in nekl-3 mutants.

We thank the Reviewer for highlighting this critical point. Indeed, *nekl-3* null mutants are inviable due to essential mitotic roles (Barstead et al., 2012, PMID: 23173093), precluding direct analysis of ciliary phenotypes. To bypass this limitation, we recently generated *nekl-3* conditional knockouts (cKOs) in ciliated neurons (Huang et al., JCB, 2025 in press, attached). In these mutants, OSM-3—which is normally enriched in the ciliary distal segment—becomes uniformly distributed along the cilium. This redistribution correlates with premature activation of OSM-3-driven anterograde motility in the ciliary middle region, consistent with our proposed model where NEKL3 phosphorylation suppresses OSM-3 activity. We have now integrated this result and discussion into the revised manuscript, reinforcing the physiological relevance of NEKL-3-mediated regulation in ciliary transport.

Page #6 line #10

“… While *nekl-3* null mutants are inviable due to essential mitotic roles (Barstead et al., 2012, PMID: 23173093), conditional knockout (cKO) of *nekl-3* in ciliated neurons (Huang et al., JCB, 2025 in press, attached) revealed its critical role in regulating OSM3 dynamics. In *nekl-3* cKO animals, OSM-3—normally enriched in the ciliary distal segment—redistributed uniformly along the cilium, concomitant with premature activation of anterograde motility in the middle ciliary region. This phenotype aligns with our model wherein NEKL-3 phosphorylation suppresses OSM-3 activity, ensuring spatiotemporal regulation of IFT.…”

(4) The methods section lacks some information, which is critical to reproducing this study.

We have now provided detailed information in the methods section in the revised manuscript.

(a) It is not described how the authors determined phosphorylation of OSM-3 by NEKL-3. In methods, nothing is described about the assay.

We performed in vitro phosphorylation assays using recombinant OSM-3 and NEKL3 purified from bacteria. We then used LC-MS/MS for identification of phosphorylation sites. We have now updated the methods section to include all the information.

Page #4 line #26

“... To determine whether NIMA kinase family members could directly phosphorylate OSM-3, we purified prokaryotic recombinant *C. elegans* NEKL-3/NEKL-4 and OSM3 protein in order to perform in vitro phosphorylation assays. We were able to obtain active recombinant NEKL-3 but not NEKL-4. The in vitro phosphorylation assays showed that NEKL-3, directly phosphorylates OSM-3 (Fig. 1A-B, Appendix Table S1). Subsequent mass spectrometric analysis revealed phosphorylation at residues 487-490, which localize to the conserved "YSTT" motif within OSM-3’s C-terminal tail region...”

Page #36, line #19

“In vitro phosphorylation assay 20 μM purified OSM-3 was incubated with 1 μM GST-NEKL-3 at 30 °C in 100 μL reaction buffer (50 mM Tris-HCl pH 8.0, 10 mM MgCl2, 150 mM NaCl, and 2 mM ATP) for 30 min. The reaction was terminated by boiling for 5 min with an SDS-sample buffer.

Mass spectrometry

Following NEKL-3 treatment, OSM-3 proteins were resolved by SDS-PAGE and visualized with Coomassie Brilliant Blue staining. Protein bands corresponding to OSM-3 were excised and subjected to digestion using the following protocol: reduction with 5 mM TCEP at 56°C for 30 min; alkylation with 10 mM iodoacetamide in darkness for 45 min at room temperature, and tryptic digestion at 37°C overnight with a 1:20 enzyme-to-protein ratio. The resulting peptides were subjected to mass spectrometry analysis. Briefly, the peptides were analyzed using an UltiMate 3000 RSLCnano system coupled to an Orbitrap Fusion Lumos mass spectrometer (Thermo Fisher Scientific). We applied an in-house proteome discovery searching algorithm to search the MS/MS data against the *C. elegans* database. Phosphorylation sites were determined using PhosphoRS algorithm with manual validation of MS/MS spectra.”

(b) The method of structural prediction by Alfafold2 and LocalColabFold needs clarification. In general, the prediction gives several candidates. How did the authors choose one of these candidates?

We generated five candidate models and all of them showed similar conformation. We thus chose the model with the highest confidence. We have provided PAE and pLDDT as additional data in Figure S2 and discussed them in the revised text on, Page #4, line #32:

“...To gain structural insights from this motif, we employed LocalColabFold based on AlphaFold2 to predict the dimeric structure of OSM-3 (Evans et al., 2022; Jumper et al., 2021; Mirdita et al., 2022). The highest-confidence model was selected for further analysis (Fig. 1C, Fig. S2)...”

(c) The methods to predict conformational changes by introducing various point mutations are interesting (Figure 3). However, the methods require more detailed descriptions. In the current form, the manuscript only lists the tools used. The pipelines and parameters need to be described. This information is important because AlphaFoldbased predictions often give folded conformations because the training data are mainly composed of folded proteins. It is surprising that the methods applied here give open conformations induced by point mutations.

We have described the pipelines in the revised Methods section on page#34, line#25:

“…OSM-3 model was predicted using LocalColabFold (Evans et al., 2022; Jumper et al., 2021; Mirdita et al., 2022). Mutated proteins were designed by Pymol 2.6, choosing the rotamer of the mutated residues in G444E, PM and PD models with the least clash as the initial conformation. To predict mutation-induced conformational changes, the initial models were subjected to Pyrosetta (Chaudhury et al., 2010). The energies of pre-relaxed models were evaluated with Rosetta Energy Function 2015 (Alford et al., 2017), and then the relax procedure were applied to the models with default parameters to obtain the relaxed models visualized by Pymol to minimize the energy of these models. In detail, to obtain the relaxed models visualized by Pymol and minimize the energy of these models, the classic relax mover was used in the procedure mentioned above with default settings. The relax script has been uploaded to Github: https://github.com/young55775/RosettaRelax_for_OSM3

(5) The authors have purified proteins. Do they show different properties in gel filtration that are consistent with the structural prediction? It is anticipated that open-form mutants are eluted from earlier than closed forms.

We thank the reviewer for this insightful suggestion. Indeed, our recent study supported that the open-from of the active OSM-3 G444E mutation were eluted earlier than the wild-type closed form (Xie et al., EMBO J., 2024). While the current study did not perform gel filtration chromatography (SEC) to directly compare the hydrodynamic properties of the OSM-3 mutants, our functional assays provide robust evidence for conformational changes predicted by structural modeling. For example: ATPase activity assays revealed that the open-state mutants (e.g., G444E and PD muatnts) exhibited significantly enhanced enzymatic activity (Figure 4A), consistent with structural predictions of an active, destabilized autoinhibitory interface (Figure 3A). These functional readouts collectively validate the predicted structural states. While SEC could further corroborate these findings by distinguishing compact (closed) versus extended (open) conformations, we prioritized assays that directly link structural predictions to in vitro enzymatic activity and in vivo ciliary transport dynamics. Future studies incorporating SEC or cryo-EM will provide additional biophysical validation of these states.

We have revised the text in the manuscript (Page #7, Lines #22):

“…Notably, the open-state OSM-3 mutants (e.g., G444E) displayed elevated ATPase activity, consistent with structural predictions of autoinhibition release (Fig. 3A, Fig. 4A) (Xie et al., 2024). While hydrodynamic profiling (e.g., SEC) could further resolve conformational states, our functional assays directly connect predicted structural changes to altered biochemical and cellular activity...”

Minor point(1) Line 85 "MIMA kinase family" should be "NIMA kinase family".

We have corrected the typo and appreciate that the Reviewer for pointing it out.

(2) M.S. and D.S. need to be defined in Figure 2D.

We have updated the figures.